# Patient Heterogeneity in Acute Myeloid Leukemia: Leukemic Cell Communication by Release of Soluble Mediators and Its Effects on Mesenchymal Stem Cells

**DOI:** 10.3390/diseases9040074

**Published:** 2021-10-16

**Authors:** Elise Aasebø, Annette K. Brenner, Maria Hernandez-Valladares, Even Birkeland, Olav Mjaavatten, Håkon Reikvam, Frode Selheim, Frode S. Berven, Øystein Bruserud

**Affiliations:** 1Department of Clinical Science, University of Bergen, 5020 Bergen, Norway; elise.aasebo@uib.no (E.A.); annette.brenner@uib.no (A.K.B.); hakon.reikvam@uib.no (H.R.); 2The Proteomics Facility of the University of Bergen (PROBE), University of Bergen, 5020 Bergen, Norway; maria.hernandez-valladares@uib.no (M.H.-V.); even.birkeland@uib.no (E.B.); olav.mjaavatten@uib.no (O.M.); Frode.Selheim@uib.no (F.S.); Frode.Berven@uib.no (F.S.B.); 3Department of Medicine, Haukeland University Hospital, 5021 Bergen, Norway

**Keywords:** acute myeloid leukemia, mesenchymal stem cells, proteomics, conditioned medium, chemokine, vesicle, intracellular transport, cell communication

## Abstract

Acute myeloid leukemia (AML) is an aggressive bone marrow malignancy, and non-leukemic stromal cells (including mesenchymal stem cells, MSCs) are involved in leukemogenesis and show AML-supporting effects. We investigated how constitutive extracellular mediator release by primary human AML cells alters proteomic profiles of normal bone marrow MSCs. An average of 6814 proteins (range 6493−6918 proteins) were quantified for 41 MSC cultures supplemented with AML-cell conditioned medium, whereas an average of 6715 proteins (range 6703−6722) were quantified for untreated control MSCs. The AML effect on global MSC proteomic profiles varied between patients. Hierarchical clustering analysis identified 10 patients (5/10 secondary AML) showing more extensive AML-effects on the MSC proteome, whereas the other 31 patients clustered together with the untreated control MSCs and showed less extensive AML-induced effects. These two patient subsets differed especially with regard to MSC levels of extracellular matrix and mitochondrial/metabolic regulatory proteins. Less than 10% of MSC proteins were significantly altered by the exposure to AML-conditioned media; 301 proteins could only be quantified after exposure to conditioned medium and 201 additional proteins were significantly altered compared with the levels in control samples (153 increased, 48 decreased). The AML-modulated MSC proteins formed several interacting networks mainly reflecting intracellular organellar structure/trafficking but also extracellular matrix/cytokine signaling, and a single small network reflecting altered DNA replication. Our results suggest that targeting of intracellular trafficking and/or intercellular communication is a possible therapeutic strategy in AML.

## 1. Introduction

Acute myeloid leukemia (AML) is an aggressive and heterogeneous malignancy characterized by accumulation of immature leukemic cells in the bone marrow [1,2]. The promyelocytic variant is characterized by specific genetic abnormalities, different treatment and better prognosis compared with the other variants [3], and in the present article the term AML refers to the non-promyelocytic variants of the disease. The AML cell population has a hierarchical organization [4], and disease development is supported by various non-leukemic bone marrow cells [5,6]. The minority of leukemic stem cells is then localized to specialized bone marrow stem cell niches that are formed by various nonleukemic cell subsets, including mesenchymal stem cells (MSCs) [6].

The crosstalk between the AML cells and the AML-supporting cells is mediated both by direct cell–cell contact and by the constitutive release of soluble mediators [5]. The leukemic stem cells constitute a small minority of the hierarchically organized AML cell population, and the majority of more mature leukemic cells thereby has a more important influence on the extracellular bone marrow microenvironment. Previous studies suggest that high constitutive cytokine release is associated with a favorable prognosis [7]. More recent studies have also shown that AML patients are not only heterogeneous with regard to the constitutive cytokine release by the leukemic cells but also in their release of a wide range of other diverse proteins derived from various cellular compartments [8,9]. However, MSCs also show constitutive release of several protein mediators that are important for leukemogenesis and probably also for chemosensitivity of human AML cells. [8,9]. Thus, there is a bidirectional crosstalk between MSCs and AML cells, and this communication both supports the leukemic cells and modulates the MSCs [5,6,10,11,12,13,14,15].

The molecular crosstalk between MSCs and AML cells also modulates the local cytokine network of the common bone marrow microenvironment [16]. The crosstalk is associated with altered mRNA levels of several cytokines/chemokines in the MSCs [16]. The aim of the present study was therefore to further investigate at the protein levels (i) whether AML cells have effects on MSCs that differ between patient subsets; and (ii) whether the previously described common AML-associated effects on the mRNA profile of MSCs can be verified at the protein levels.

## 2. Materials and Methods

### 2.1. Preparation of AML Cells and MSCs

The AML cells were collected after written informed consent and in accordance with the Declaration of Helsinki. The Regional Ethics Committee approved both the collection of biological material in the biobank (REK Vest 2015/1759; approved 05.11.2015) and the use of the cells in the present study (REK Vest 2017/305, approved 07.04.2017). The clinical and biological characteristics are summarized in Table 1 and are presented in detail in Appendix A. 

All patients had a high percentage of circulating AML cells (>80% among peripheral blood leukocytes) and highly enriched AML cell populations (>95% leukemic cells, only a minor subset of contaminating small lymphocytes) could thereby be prepared by density gradient separation alone. The patients are consecutive group, i.e., they represent an unselected group of patients fulfilling the criteria of high percentage of circulating AML cells and derived from a defined geographical area during a defined time period.

The experimental model for the present study has been described in detail previously [7,9,13,14,15,16]:AML conditioned medium. AML cells were cultured (1 × 10^6^ cells/mL, 10 mL medium per flask) in T25 flasks (Falcon; Glendale, AZ, USA). The culture medium was serum-free IMDM without phenol red (Ref. 21056023, Thermo Scientific; Waltham, MA, USA). Supernatants were collected after 48 h and stored in aliquots at −80 °C until used. The supernatants are referred to as AML conditioned medium or AML-CM. Supernatants were harvested after 48 h as described in several previous studies [7,9,13,14]; differences between patients can then be detected and these differences are associated with clinical chemosensitivity and differences in overall survival after intensive therapy [7].MSC expansion. We investigated AML effects on normal MSCs derived from a healthy donor; these MSCs were used instead of autologous MSCs to have a similar system for evaluation of direct AML effect for all patients without any influence of indirect AML effects mediated for example through other stromal cell subsets. The MSCs from a healthy donor (C-12974, lot number 427Z010.1; Promocell Gmbh, Heidelberg, Germany) were thawed according to the manufacturer’s instructions, and 5 × 10^5^ cells were expanded to 4 × 10^6^ cells in Mesenchymal Stem Cell Growth Medium (Promocell Gmbh) before the cells were distributed into four T75 flasks (Falcon) after eight days of culture. The cells were cultured for three additional days before the medium was changed to serum-free IMDM. The cells were then transferred to 24-well culture plates (Falcon; 3 × 10^4^ cells/well) and cultured in Mesenchymal Stem Cell Growth Medium for one additional day before the medium again was changed to serum-free IMDM.MSC cultures with AML conditioned medium. AML-CM (also prepared in serum-free IMDM) was added at a ratio of 1:1 (24 well culture plates, a total of 2 mL medium/well). MSCs were cultured in the presence of AML-CM for 48 h. Four control replicates of MSCs were also cultured with culture medium instead of AML-CM. MSCs from the same donor were used in all cultures. After the culture period MSCs were harvested and lysed in 150 µL lysis buffer (4% SDS/0.1M Tris-HCl, pH 7.6) as described previously [17] and stored in aliquots at −80 °C until analyzed. As explained above cells were cultured in serum-free medium during the exposure to AML conditioned medium and not in the presence of specialized growth medium. Previous studies have shown that MSC proliferation is maintained during a 48-h culture period [15,16], and light microscopy showed that the high MSC viability was maintained during culture.

### 2.2. Proteomics Sample Preparation

For protein digestion of 5–10 µg protein samples the single-pot, solid-phase-enhanced sample preparation SP3 method was used [18,19], but with the addition of three extra washes with 80% EtOH for SDS removal and sample tube exchange after the third and the sixth washing step. NanoDrop UV-Vis spectrophotometer (Thermo Scientific, Waltham, MA, USA) was used to measure peptide concentration prior to LC-MS/MS analysis.

### 2.3. Liquid Chromatography (LC) Tandem Mass Spectrometry (MS) Analysis

Samples containing 0.6 µg tryptic peptides dissolved in 2% acetonitrile (ACN) and 0.5% formic acid (FA) were injected into an Ultimate 3000 RSLC system (Thermo Scientific, Waltham, MA, USA) online coupled to an Orbitrap Eclipse Tribrid mass spectrometer equipped with an EASY-IC/ETD/PTCR ion source and FAIMS Pro interface (Thermo Scientific, San Jose, CA, USA). The LC and column specifications have been described previously [8], except that the gradient composition, using 100% ACN as solvent B, was as follows: trapping over 5 min with 5% B followed by 5–7% B for 1 min, 7–22% B for the next 129 min, 22–28% B for 14 min, and 28–80% B for 7 min, hold 80% B for 18 min and ramp to 5% B for 6 min. To avoid sample carryover effect, a trap and valve wash run of 15 min was included after each sample.

Instrument control was through Tune 3.4.3072.18 and Xcalibur 4.4. The MS1 resolution was 120,000 and the scan range 375–1500 m/z, AGC target was set to standard, maximum injection time was automatic and RF lens at 30%. The intensity threshold at 5.0 × 10^4^ and dynamic exclusion lasted for 30 s. The MS/MS scans consisted of HCD with collision energy at 30%, quadrupole isolation window at 4 m/z and Orbitrap resolution at 15,000. FAIMS was set up with the standard resolution mode and a total gas flow of 3.6 L/min. The CVs were set to −45, −65 and −80 V. As a quality control for the LC-MS system, a control sample from HeLa cell culture was run in the beginning, during and in the end of the sample sequence (data not shown).

### 2.4. Statistical and Bioinformatical Analyses

The raw LC-MS files were searched in the Proteome Discoverer Software (version 2.5, Thermo Fisher Scientific, Bremen, Germany) using the SEQUEST HT database search engine with Percolator validation (FDR < 0.01), against the concatenated reviewed Swiss-Prot human database (canonical and isoform FASTA, downloaded from Uniprot 30.04.2021). Oxidation (M), Gln->pyro-Glu (N-terminal peptides) and acetylation (N-terminal proteins) were set as dynamic modifications and carbamidomethylation (C) was set as fixed modification. Default settings were used, except that total peptide amount for all peptides was used for normalization, and summed abundances were used for protein abundance calculations.

The normalized protein abundances were imported into the Perseus software (version 1.6.1.1, Max Planck Institute for Biochemistry, Martinsread, Germany) [20] for data filtering, processing and analysis (contaminants and proteins with medium or low confidence were removed from the data set). Proteins with at least three valid values (normalized protein abundance value) in each group were selected for statistical comparisons using Welch’s *t*-test and *Z*-statistics [21]. *p*-values < 0.01 were considered significant.

Perseus was used for hierarchical clustering with Pearson correlations as distance metrics and complete linkage. Gene ontology (GO) analyses were performed using a GO tool [22] in “compare_samples” mode, where the GO slim subset according to Generic GO slim by GO consortium were reported and GO terms with FDR < 0.05 were considered significantly enriched. Significantly regulated proteins and proteins only quantified in MSCs cultured with AML conditioned medium were imported to the STRING database (version 11.0 [23]) using experiments and databases as interaction sources and 0.7 as minimum required interaction score (i.e., high confidence). Cytoscape (version 3.3.0 and 3.8.2; National Institute of General Medical Sciences, Bethesda, MD, USA) and MCODE (version 2) were used to visualize and classify highly connected protein cluster [24,25].

## 3. Results

### 3.1. AML Cells Have Limited Effects on the MSC Proteoma; Most MSC Protein Levels Are Not Altered by AML Conditioned Medium

MSCs derived from the same healthy donor were cultured with AML conditioned medium derived from 41 patients; four additional and independent control cultures were also prepared in medium alone (i.e., referred to as MSC medium controls). An average of 6814 proteins (range 6493–6918 proteins) were quantified in the 41 MSC samples derived from cultures prepared with the AML conditioned media, whereas an average of 6715 proteins (range 6703–6722) were quantified in the four medium controls. 

Correlation analyses of all the individual samples based on the normalized protein abundances showed a Pearson R correlation between 0.881 to 0.986 (average 0.968) (Figure 1, left), and the large majority of proteins were expressed at detectable levels both when MSCs were cultured with the conditioned media and in the four medium control cultures (Figure 1, right). Thus, qualitative analyses of MSC proteomic profiles showed extensive similarities both when comparing individual patient conditioned medium samples and when comparing patient and control samples. However, three exceptional patients (Figure 1 left; corresponding to the three upper rows/left columns) deviated from the other ones, and these patients correspond to patients 8–10 in Appendix A.

The correlation analyses for the four MSC medium controls are presented in detail in Appendix A. Thus, the control MSCs showed a very small variation range with regard to quantified proteins, and this correlation analysis illustrates that variation in the levels of individual proteins between control samples is small.

### 3.2. The Effects on the MSC Proteomic Profile by Constitutive AML Cell Mediator Release Differ between Patients

We did an unsupervised hierarchical cluster analysis of our overall results based on the 6882 proteins that were quantified for at least 23 of the 45 samples (Figure 2, 41 patient and four control samples). The MSC samples clustered into two main clusters/subsets (indicated by yellow and brown to the right in the figure), and the larger brown main subset could be further divided into two subclusters (indicated by light and dark brown in the figure). Furthermore, we identified two main protein clusters (indicated by red and green, see the top of the figure), and each of these two main clusters could be further subdivided into two subgroups.

As described in Section 3.1, we identified three exceptional patients in the correlation analysis presented in Figure 1 (left part). These three patients are referred to as patients 8–10 in Appendix A; they are the three lower patients in the upper/yellow main patient cluster in Figure 2 where they form a separate subset/subcluster among the 10 upper main cluster patients. All these three exceptional patients 8–10 were elderly males with secondary AML, but they differed with regard to differentiation (i.e., FAB classification, CD34 expression) and genetic abnormalities (Appendix A).

The hierarchical clustering analysis presented in Figure 2 showed that the protein profiles for most MSC samples had many similarities, i.e., 31 patient samples were included in the lower brown/light brown cluster together with the four MSC control samples that were only exposed to medium alone but not AML-conditioned medium. This last observation suggests that the conditioned medium induced relatively small MSC proteomic alterations in these 31 MSC samples because they clustered together with the four unexposed MSC controls. The last 10 patient samples differed from the other patents and clustered together in the upper yellow patient cluster. This upper/yellow main patient cluster (i.e., patients 1–10 in Appendix A) had a significantly higher frequency of patients with AML following previous myelodysplastic syndrome or chronic myeloproliferative neoplasia (Fisher’s exact test, *p* = 0.0165) than the other patients (Patients 11–41), but these two main clusters/patient subsets did not differ with regard to patient age, sex, AML cell differentiation (morphology/FAB, expression of the CD34 stem cell marker) or genetic abnormalities (karyotype, *FLT3/NPM1* abnormalities).

### 3.3. The Constitutive Mediator Release by Primary AML Cells Derived from a Minority of Patients Modulates Mitochondrial Metabolism and Protein Metabolism/Extracellular Matrix Protein Release by Normal MSCs

We compared the levels of individual proteins for the two main patient subsets identified in Figure 2, i.e., the upper 10 patients in the yellow main cluster versus the lower brown main cluster including the other 31 patients. The first analyses only included proteins that could be quantified for at least three patients in each of these two main patient subsets, and a significant difference was then defined as a *p*-value < 0.01 (both for the t-test and for the Z-statistics of fold changes). A significant difference was observed for 217 MSC proteins according to these criteria; 150 proteins were significantly higher for the 10 patients in the upper yellow and 67 proteins significantly lower in the yellow subset patients compared with the 31 patients in the other brown/light brown main cluster (Appendix A).

We did a GO analysis based on the 217 identified proteins that differed significantly between the two main patient clusters (Table 2). The upper yellow patient cluster showed increased levels of proteins involved in mitochondrial function/metabolic regulation, whereas the lower brown main cluster showed increased levels of (glyco)proteins involved or included in secretion/extracellular matrix/protein modulation. Thus, the AML-derived and constitutively released mediators differ between the two main patient subsets with regard to their overall effect on normal MCs; the smaller upper/yellow patient subset is characterized by upregulation of several proteins involved in mitochondrial functions/cell metabolism and downregulation of proteins involved in protein metabolism/extracellular release/extracellular matrix compared with the majority of 31 patients that showed less extensive effects and clustered together with the normal MSC controls in the lower brown/light brown patient cluster.

We thereafter conducted a network analysis based on the 217 proteins identified in the patient cluster comparison (see above). We first conducted a protein–protein interaction network analysis in String based on the 217 MSC proteins that differed significantly between the two patient main clusters/subsets (see Figure 2); the results from this analysis are presented in Appendix A. Furthermore, we identified subclusters by using the MCODE application in Cytoscape; the results from this last network analyses also reflected differences in cellular communication (Figure 3) similar to the observations from the GO analysis presented in Table 2.

We finally identified MSC proteins that were quantified mainly for one of the two patient clusters: (i) for at least 5 of the 10 patients in the upper yellow cluster but only 3 or less of the 31 patients in the brown lower cluster; or (ii) for at least 15 of the 31 patients in the brown cluster but for three or less patients in the yellow cluster. Only 51 proteins fulfilled these criteria; 38 proteins were expressed mainly for patients in the lower brown main cluster whereas 13 proteins showed increased levels for the 10 patients in the upper yellow cluster. These 51 proteins are listed in Appendix A. This analysis also identified individual proteins that are involved in cellular communication, whereas relatively few of these proteins were involved in metabolic regulation.

### 3.4. Comparison of the AML Secretome for the Two Main Patient Subsets Identified in the Unsupervised Hierarchical Clustering Analysis

The AML secretome for 40 of the 41 patients was characterized in a previous study [9], and based on these previously published results we compared the AML secretome for the two main patient clusters identified in our unsupervised hierarchical cluster analysis (Figure 2). We first did an analysis based on soluble proteins that could be quantified for at least three patients in each of these two main patient subsets, and a significant difference was then defined as a *p*-value < 0.01 both for the t-test and for the Z-statistics of fold changes. This comparison identified only 22 proteins; 11 proteins showed higher levels for the yellow upper patient cluster and 11 proteins showed higher levels in the lower brown cluster (Appendix A). This analysis reflected a difference between these two patient subsets with regard to constitutive extracellular AML cell release of soluble protein mediators involved in cellular communication.

We next compared soluble protein mediators that were quantified mainly for one of the two main patient clusters: (i) for at least half/5 of the 10 patients in the upper yellow cluster but only for 3 or less of the 31 patients in the brown lower cluster (10 proteins identified); or (ii) for at least half/15 of the 30 patients from the brown cluster but for three or less patients in the yellow cluster (67 proteins identified) (Appendix A). This comparison also showed that the two main patients subsets identified in Figure 2 differed with regard to the constitutive extracellular release by the AML cells of proteins involved in or being important for cellular secretion/extracellular matrix (Appendix A), and the overview of the classification of these proteins (Appendix A) shows that they are involved in both vesicle formation, cytoskeletal function, intracellular trafficking/endocytosis, cell–cell adhesion, cell–matrix adhesion and modulation of extracellular matrix molecules.

To summarize, we used two different approaches to identify constitutively released AML cell proteins that differed between the two main patient subsets identified in the hierarchical cluster analysis. Based on these two strategies we conclude that the constitutive AML secretomes for the two main patient subsets identified in the MSC clustering analysis (Figure 2) reflect differences in the secretory mechanisms and in addition differences in the levels of matrix molecules or extracellular proteins directly involved in cell-cell communication (e.g., matrix molecules, cytokine, cytokine receptors, soluble glycoproteins/adhesion molecules).

### 3.5. A Subset of MSC Proteins Can Only Be Quantified after Exposure to the AML Secretome but Even These Proteins Contribute to AML Heterogeneity

We identified 301 proteins (see Appendix A for the complete list) that were only quantified for MSCs exposed to AML conditioned medium but not for any of the control MSCs cultured in medium alone. However, we would emphasize that these observations should be interpreted with great care because only four control MSCs could be included in the comparison. Only 30 of these 301 AML-induced proteins were detected for at least 30 of the 41 samples/patients (Appendix A; see also see Appendix A for more detailed descriptions of individual proteins), but more than half of these proteins were detected for less than half of the patients. Finally, no proteins were quantified only for control MSCs but not for any of the MSCs exposed to AML conditioned medium.

The number of proteins that reached quantifiable levels only after MSC-exposure to AML conditioned medium, varied considerably between the individual patients (Appendix A). None of the MSC samples expressed all these proteins (median number 90 proteins, range 34–174 proteins), however, the 10 samples belonging to the upper yellow patient cluster identified in Figure 2 expressed significantly higher levels of these AML-induced proteins (median 128, range 78–174) than the other 31 samples (median 81 proteins, range 34–131 proteins, Mann–Whitney U test, *p* = 0.00036). This observation is consistent with our previous conclusion based on the hierarchical clustering analysis (Figure 2); the larger lower patient cluster included all four control MSCs suggesting that the AML effects on the MSC proteome were relatively weak, whereas the AML cells derived from the 10 patients in the upper patient cluster have more extensive effects on the MSC proteome including expression of a higher number of AML-induced proteins (Appendix A).

We conducted protein–protein interaction network analyses (using the String database) based on the proteins only quantified for MSCs exposed to the AML conditioned media (Appendix A). The protein network based on all 301 proteins is presented in Appendix A whereas Appendix A present the networks based on the 30 proteins detected for at least 30 patients and the 73 proteins detected for at least 20 patient samples, respectively. An additional analysis of the overall protein network using the MCODE application in Cytoscape identified three subclusters of densely connected proteins from the original network (Figure 4). We did a further classification of the AML-induced MSC proteins for each of these networks by using a GO tool (Table 3). We found that the AML-induced network proteins were important especially for plasma membrane/extracellular region/space (Figure 4, all three networks), cytoplasmic vesicles/vesicle-mediated transport (Figure 4, networks 1 and 2) and cytokine/chemokine-mediated signaling (Figure 4, networks 2 and 3).

The characteristics of the 30 individual AML-induced proteins that could be quantified for at least 30 patients (listed and described more in detail in Appendix A) can be summarized in the following way:A large subset of these 30 proteins are important for the biological functions of GTP (e.g., GTPases, G-proteins, GTPase regulated protein) including VAV1, CXCL7/PPBP, AGTPBP1, RAB37, DEFA1, PLCB2, ARHGAP15/25/45, GIMAP1/8 and DOK2. Several other CCL/CXCL chemokines were quantified in smaller subsets of patients, including CCL5, CCL24, CXCL1, CXCL3, CXCL4/PF4, while CXCL8 was quantified for most patients (Figure 4).Several of the GTP associated proteins regulate Rho activity and thereby also vesicular biology/trafficking, e.g., VAV1, UNC13D, RAB37, CD36 and DEFA1.Several of the 30 proteins listed in Appendix A are important for the cytoskeleton/actin (VAV1, BIN2, ARHGAP15), cell signaling (TLR2, IRAK3, LYN, ICAM3, SASH3, CXCL7/PPBP) and cell adhesion/extracellular matrix (CD84, ICAM3, CD36, possibly also the proteases GZMK, CFD).No proteins reflected differentiation induction of the MSCs, but two proteins have been described to maintain stemness (BIN2, CD36).No proteins are known as cell cycle regulators; this is consistent with previous studies describing only minor or no effects of AML cells on MSC proliferation [9,16].None of the 30 proteins were among the 100 top-ranked exosomal proteins (ExoCarta: Exosome markers, analyzed 20.06.2021).

Thus, regulation of vesicle biology/trafficking and modulation of the extracellular microenvironment seems to be the most important functions of the AML-induced proteins quantified for most MSCs cultured with conditioned medium from AML patients. Even though these results should be interpreted with care because we had only four control MSCs, we would emphasize that the observations are similar to the results from the comparison of the two main patient subsets identified in Figure 2.

We classified these 301 altered proteins by using a GO tool; the most significant terms are presented in Appendix A. Even though we would emphasize that these results should be interpreted with care because we had only four control samples, we would emphasize that we again observed differences in terms that reflect extracellular protein release, i.e., upregulated terms reflecting extracellular space/cytoplasmic vesicles and downregulated terms reflecting differences in extracellular matrix.

### 3.6. Common Effects of AML Cells on the MSC Proteome: Comparison of Protein Abundances for Proteins Quantified Both in Control MSCs and after MSC Exposure to AML Conditioned Medium

We compared the protein abundances of individual proteins for MSCs incubated with AML conditioned medium and control MSCs incubated in medium alone. The analyses only included proteins that could be quantified for at least three of the four control MSC samples, and a significant difference was defined as a *p*-value < 0.01 (both for the *t*-test and for the Z-statistics of fold changes). A total of 201 proteins could be classified as significantly altered according to these criteria; 148 proteins were upregulated after exposure to AML conditioned medium whereas 53 were downregulated (see Appendix A for the complete protein list). All these differently expressed proteins were quantified for at least 24 of the 41 patients.

With regard to individual proteins (Appendix A), we would emphasize the following:A relatively large group of mitochondrial proteins showed altered levels after exposure to AML conditioned medium but were not included in any of the large protein networks: ACSS1, CYBB, CLYBL, ACADS, COX7A2L, MGME1, COQ8A, CYP27A1, MCAT, COX7A2, ETFDH, AGMAT, DGUOK, ATP5MF, PCCB, TRMU, ECH1, SOD2, PGS1 and MTERF3. All except the last protein showed increased levels after conditioned medium exposure.Some proteins important for the biological functions of GTP (ARHGDIB, RAB3A, MX1, IQGAP), adhesion molecules (the integrins ITGAM, ITGB2, ITGAX together with ICAM1 and VCAM1) and S100 molecules (S100A9, S100A8, S100A12) were also increased after exposure.The three collagens COL6A2, COL4A2 and COL5A1 showed decreased levels after exposure to AML conditioned medium.Several kinases (CAMK4, NME7, CDK2, COQ8A, DGUOR, WNK4, CDKNA1, PFKFB3, LATS2) and phosphatases (PTPRJ, PTPN6, INPP5A, G6PC3, PTPRE) showed altered levels after exposure; all except the three last kinases then showing increased levels. These observations show that several proteins involved in the regulation of posttranscriptional protein modification were altered.With regard to cytokine biology no chemokines showed altered levels. However, IL18 and IL17RA showed increased levels, whereas decreased levels were seen for VEGFC, IGFBP3 and IGF2.LOX, LOXL1 and LOXL2 showed increased levels after exposure.Only two of the proteins (LGALS3BP, STOM) were among the 100 top-ranked exosomal proteins (ExoCarta: Exosome markers, analyzed 20.06.2021).

These alterations of protein levels further illustrate that even though conditioned medium exposure had significant effects on relatively few proteins (<10% of the quantified proteins), these proteins are involved in fundamental cellular functions. Thus, our overall results, including the comparisons with the four control MSCs that require careful interpretation, suggest that a major effect of AML-derived soluble mediators on the MSC proteome is altered regulation in the release of soluble mediators.

We did a network analysis based on all 201 proteins (Appendix A). An additional analysis using the MCODE application in Cytoscape identified four networks/subclusters that included at least five densely connected proteins (Figure 5). We did a further classification of the AML-modulated/associated MSC proteins for each of these four networks/subclusters by using a GO tool (Appendix A). The largest detected network reflected increased levels of proteins important for secretory granule/extracellular space/cytoplasmic vesicle lumen (cluster 1) in the MSCs exposed to AML conditioned medium, whereas three smaller clusters reflected altered cell adhesion (cluster 2), DNA replication (cluster 3) and intracellular organellar lumen (cluster 4). Thus, a large part of the AML-associated proteins also reflects differences in vesicle function/transport, and this is similar to many of the AML-induced proteins (see Section 3.3).

### 3.7. A Small Number of Proteins Could Be Detected for Control MSCs but Only for a Small Minority of MSCs Exposed to AML Conditioned Medium

A total of 6958 proteins were expressed by at least one of the four MSC control samples; all four MSC control samples expressed 6423 of these proteins and 6655 proteins were expressed by at least three of the control samples. None of the 6958 MSC proteins were only expressed by control MSCs and not by any of the AML-exposed MSCs. However, 27 proteins were quantified in MSC controls but for less than 10 of the 41 AML exposed MSC samples, and these proteins are involved in the following cellular functions (Appendix A):Extracellular matrix: COL6A3, POSTN, CHIT1, FSD1, ODAPH;Intracellular transport, cytoskeleton: KTN1, PIP, FSD1, TTC26;Transcriptional regulation/DNA binding: BUD13, ERCC6, MBD1, ZNF22;Protein synthesis/degradation/modification: KTN1, EIF1B, CAMKK2, SERPINB12, MINDY1, SENP1, TXNDC11, HSP90AB3P;Cell cycle regulation: FOLR3, S100P, CEP55;Other functions: MCC (Wnt signaling), ACOX1 (fatty acid metabolism), RNLS (oxidoreductase).

Thus, most MSC quantified proteins could also be quantified for the majority of AML exposed MSC samples. However, 27 proteins were quantified for control MSCs but only for less than 10 of the 41 AML exposed MSC samples. These 27 proteins included three proteins with high expression in MSC controls (i.e., detected for all four control MSCs) and 24 additional proteins showing only low levels (i.e., quantified in less than three MSC controls) in MSC controls.

## 4. Discussion

AML is a heterogeneous disease [1,26,27,28], and in our present study we show that AML patients are heterogeneous also with regard to AML cell modulation of leukemia-supporting MSCs. The differentially expressed MSC proteins were important especially for intracellular vesicle formation/trafficking and modulation of the extracellular microenvironment. However, future studies have to investigate the in vivo effects of the modulations (i.e., they can be detected for bone marrow MSCs derived from AML patients), and whether the bidirectional in vivo crosstalk between MSCs and AML cells is important for clinical chemosensitivity/survival.

In our present study we used highly enriched MSCs derived from the same healthy individual; we had a focus on AML cell/patient heterogeneity and by using this standardized experimental model we compared the AML effects for 41 patients. This model will reflect differences in direct but distant AML effects (i.e., effects mediated by extracellular release of soluble mediators but not cell-cell contact) on MSCs without indirect effects mediated via other stromal cells, and for the same reasons we did not use autologous MSCs. In a previous study of cocultured MSCs and AML cells we showed that the extracellular protein levels of several CCL and CXCL chemokines were increased [16]. Thus, these observations from a study investigating the constitutive release of a limited number of cytokines measured by an alternative antibody-based analysis or by gene expression profiling, are consistent/similar to our present observations.

The enriched AML cells were derived from consecutive patients with a high percentage and/or absolute number of leukemic cells among circulating leukocytes; for this reason, enriched AML cell populations (i.e., >95% leukemic cells) could be prepared by a simple and highly standardized gradient separation procedure [29,30]. Our results may therefore be representative only for patient with circulating leukemic cells. However, since the level of circulating AML cells has a relatively weak prognostic impact in patients receiving intensive therapy [31,32,33,34,35], our results are possibly representative with regard to clinical chemosensitivity also for other AML patients. Finally, due to our inclusion of consecutive patients many of them were elderly or unfit and could not receive or complete potentially curative intensive treatment [1]. Thus, our patients received either intensive treatment, AML stabilizing low-toxicity treatment or only supportive care, and these subsets included too few patients for survival analyses.

Primary AML cells show constitutive release of a wide range of proteins, including cytokines/chemokines and their soluble receptors as well as soluble adhesion molecules and proteases [7,9]. This overall AML contribution to the extracellular microenvironment is probably caused mainly by the majority of more mature cells within the hierarchically organized AML cell population. For this reason, we investigated AML conditioned medium prepared by culture of the total AML cell population. Furthermore, our AML-conditioned media were derived from cell cultures incubated for 48 h, because it was then possible to detect a patient heterogeneity in the constitutive AML cell cytokine release that is associated with relapse risk [7]. Finally, only four MSC control samples were available for our present study, and for this reason we used clearly defined and very strict criteria to define AML-induced (not detected in control cultures) and AML-modulated (expressed both in control and AML-exposed MSCs but with statistically significant differences) MSC proteins. Although these observations should be interpreted with care, we will emphasize that many of the observed differences were highly significant.

The unsupervised hierarchical cluster analysis identified two main patient subsets/clusters (Figure 2). The major brown/lower patient cluster included all four AML control samples together with 31 patient samples, suggesting that the MSCs for these patients are more similar to the MSC controls than the 10 patient MSC samples included in the minor yellow/upper cluster. Our proteomic comparison showed that these two patient clusters differed significantly with regard to proteins involved in intracellular transport (e.g., vesicle and cytoskeletal proteins) and cell adhesion, and in addition the yellow minor cluster showed increased levels of several mitochondrial and/or metabolic regulatory proteins. Thus, the 10 patients in the yellow/upper cluster show more extensive alteration in the MSC proteome after AML exposure.

We identified 301 quantifiable MSC proteins that were only detected after exposure to AML conditioned medium; we refer to these proteins as AML-induced proteins. Our GO term analyses showed that these proteins were also important especially for intracellular vesicle formation/biology/transport and cytokine/chemokine-mediated signaling. As stated above present observations are consistent with previous experimental studies describing increased extracellular release of certain cytokines/chemokines during coculture of MSCs and AML cells [16]. Furthermore, protein network analyses (Figure 4) identified three networks/clusters that also reflected similar biological functions/differences as the GO-term analyses. Finally, it should be emphasized that the patients were heterogeneous with regard to these AML-induced differences (Appendix A); none of the patients showed altered levels of all 301 proteins and several patients showed detectable levels of relatively few of these proteins. Thus, only parts of the three MSC interaction networks were induced at quantifiable levels for individual patients, but the 10 patients in patient cluster 2 (Figure 2) showed a significantly higher number of quantifiable AML-induced MSC proteins than the other 31 patients (Appendix A).

In addition to the 301 AML-induced proteins we identified 201 proteins that were quantified both in AML and control cultures but were significantly altered after exposure to AML-conditioned medium (Figure 5, Appendix A). First, several of these proteins are also involved in intracellular vesicle formation/transport, an observation further supporting our hypothesis that a main effect of the crosstalk between MSCs and AML cells is to modulate the intracellular vesicle formation/transport in the MSCs and thereby also the extracellular release. Second, the levels of several chemokines were increased after exposure to AML conditioned medium; this is also consistent with previous MSCs/AML coculture studies [16]. Finally, although we identified a small network including proteins that are important for DNA replication, only a few of the differentially expressed proteins are involved in the regulation of cell cycle progression or proliferation. This is also consistent with previous experimental studies suggesting that AML cells have only minor effects on MSC proliferation [16].

A small number of proteins could be quantified for MSCs but only for a few AML-exposed MSCs (Section 3.7, Appendix A). These observations also support the hypothesis that modulation of MSC secretion is a main effect of the AML mediator release.

Some of the identified AML-induced or AML-modulated proteins are possibly of particular importance. First, the increased levels of several CCL/CXCL chemokines can be important for the interactions between MSCs and other non-leukemic stromal cells in the bone marrow microenvironment, especially for AML-induced angiogenesis [36,37]. Second, the altered expression of collagens as well as soluble integrins and other adhesion molecules (e.g., VCAM, ICAM) may be important for the direct contact between MSCs and neighboring cells and/or extracellular matrix molecules [38,39,40]. Third, several MSC molecules involved in intracellular signaling (e.g., GTPases, Src kinases) seem to be important also in AML cells [38,41,42,43], and therapeutic targeting of such molecules may therefore represent combined direct and indirect therapeutic targeting of AML cells. Finally, several LOX molecules were modulated, and these proteins seem to be important prognostic markers in various malignancies and are regarded as potential therapeutic targets [44,45,46,47,48] even in AML [49].

In a previous study we used a proteomic strategy to investigate the constitutive protein release by primary human AML cells [9], and it was therefore possible to compare the proteomic profiles of the AML secretome for the 10 patients in the smaller yellow patient cluster and 30 patients in the larger MSC-like main cluster (Figure 2). Many of the proteins that differed between the two patient clusters/subsets either reflect the process of extracellular release or they are involved in cellular adhesion and/or communication. The soluble forms of such cell surface proteins may also have biological activity [50,51]. Our present observations therefore suggest that these differences in the AML protein-secretome are responsible for the differences in the AML effects on the MSC proteomic profiles between the two main patient clusters/subsets. However, it should be emphasized that cells can release proteins through their release of exosomes that also contain metabolites and μRNA [52]. These mediators may also contribute to the AML-associated modulation of MSC proteomic profiles.

## 5. Conclusions

Our present study suggests that the AML secretome alters the MSC capacity of extracellular protein release and thereby modulates the bidirectional MSCs communication with the leukemic cells (and possibly also communication with other stromal cells). Targeting of the crosstalk between AML cells and MSCs may represent a new therapeutic strategy, but this approach may be more effective for certain patients and should therefore be considered especially for individualized (personalized) therapy [53].

## Figures and Tables

**Figure 1 diseases-09-00074-f001:**
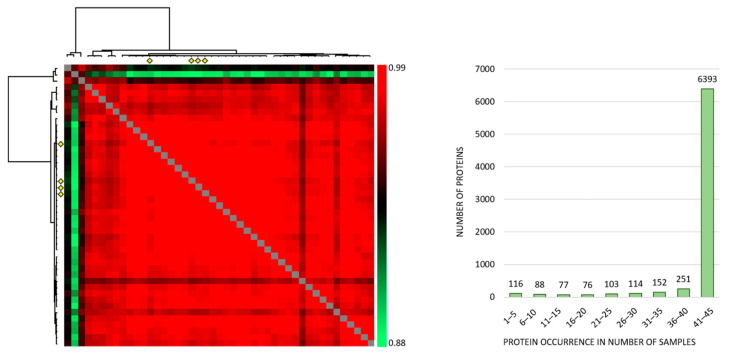
The proteomic MSC profile; an analysis of the overall results when MSCs were cultured with AML-conditioned medium (41 patients tested) and in control cultures prepared in medium alone (four independent cultures). (**Left**) The figure presents a correlation map including all 45 samples; the Pearson R correlation coefficient ranged from 0.881 to 0.986 (average 0.968). The MSC controls are indicated with four yellow diamonds (both at the x- and y-axis) and correlated well with the majority of samples. The three samples deviating most from the majority (the three upper rows/left columns) are MSCs cultured with conditioned media from AML patients 8–10 in Appendix A, respectively. (**Right**) The histogram shows the number of samples with detectable levels of each individual protein (i.e., protein occurrence); 6318 proteins were quantified in at least 41 of the 45 samples.

**Figure 2 diseases-09-00074-f002:**
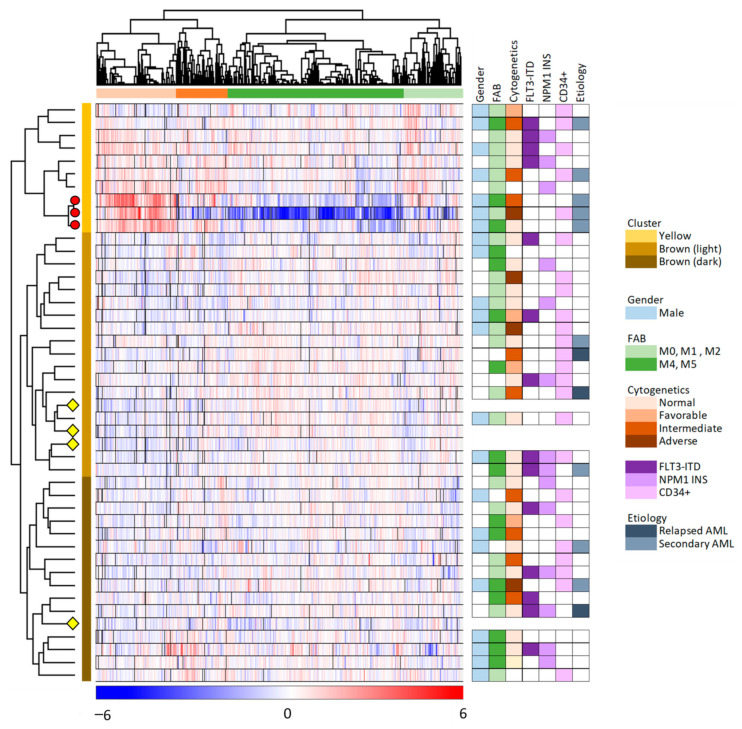
Unsupervised hierarchical clustering analysis based on the protein expression profiles of 6882 proteins which were quantified in at least 50% of the 45 MSC samples. The color key below the hierarchical cluster indicates the log_2_-transformed and Z-scored protein expression values. The MSC samples (i.e., 41 MSC samples incubated with AML conditioned medium and four MSC medium controls) clustered into two distinct main clusters/groups (yellow and brown, indicated to the right of the cluster), with subgroups (brown main group including the light and dark brown subgroups). The four MSC control samples are indicated with yellow diamonds, and the three exceptional patients identified in the left part of Figure 1 are marked with red circles. The proteins clustered into two main groups (orange and green; see top of the figure), with two subgroups each. AML patient characteristics are presented to the right in the figure and detailed in Appendix A.

**Figure 3 diseases-09-00074-f003:**
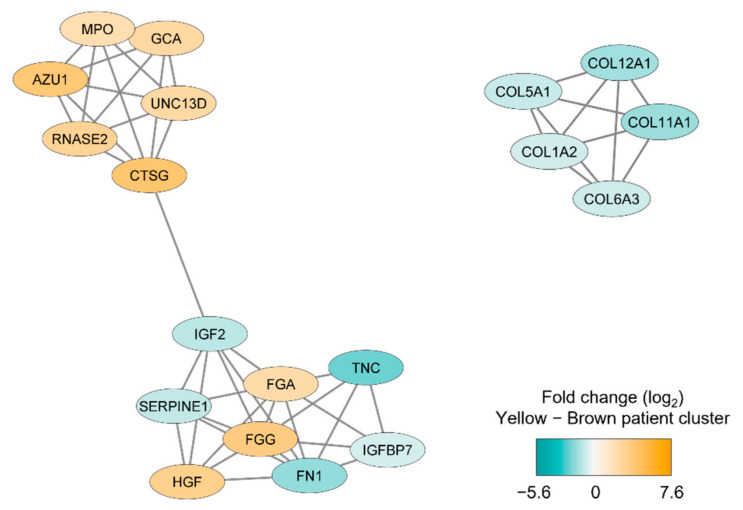
Subclusters including at least five densely connected MSC proteins that differed significantly between the two main patient clusters identified by the unsupervised hierarchical cluster analysis (Figure 2). The subclusters were identified by the MCODE application in Cytoscape from a protein-protein interaction network analysis in String based on 217 MSC lysate proteins that differed significantly between the two patient subsets. The color coding of the protein nodes indicates the log_2_-transformed protein fold change, where turquoise indicates decreased protein abundance and orange indicates increased protein abundance (i.e., increased in the brown main patient color) in the upper yellow main patient cluster (*n* = 10).

**Figure 4 diseases-09-00074-f004:**
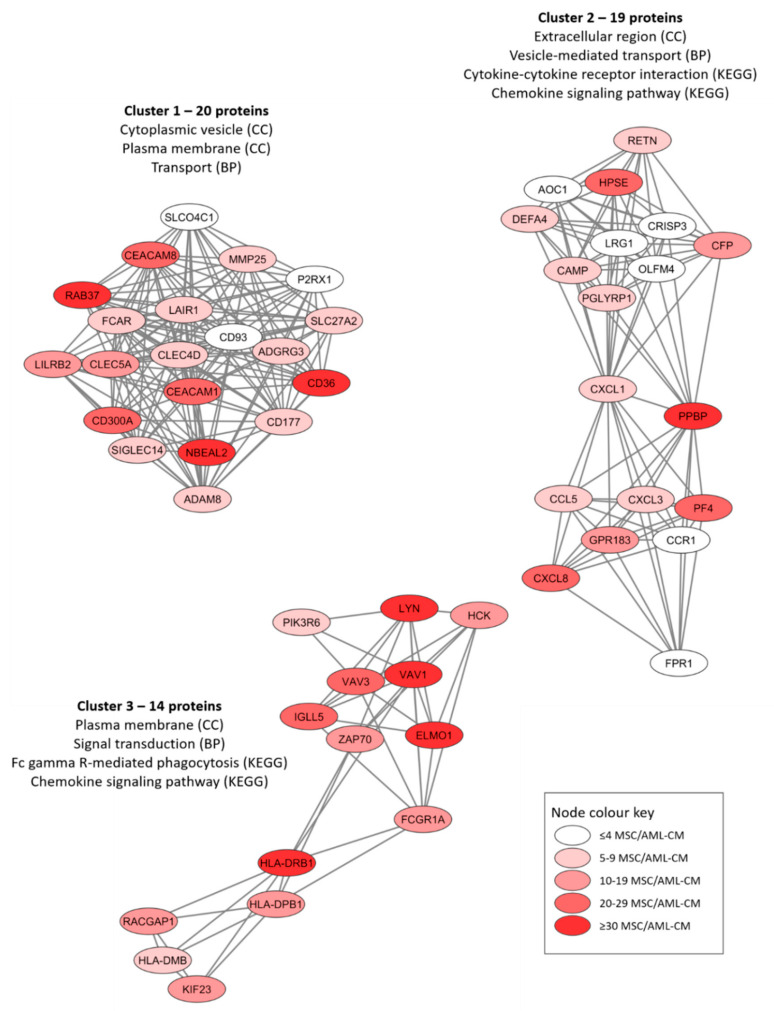
Three networks/subclusters of densely connected AML-induced MSC proteins. The networks were identified by the MCODE application in Cytoscape from a protein-protein interaction network analysis in String, and these analyses were based on 301 MSC proteins that could be quantified in MSCs only after exposure to AML conditioned medium (AML-CM) (see Appendix A for the overall network results). The figure presents the most significant GO-terms/KEGG pathways for each cluster/network, and the color coding indicates the number of patients/samples with quantifiable levels for each individual protein (see lower right box).

**Figure 5 diseases-09-00074-f005:**
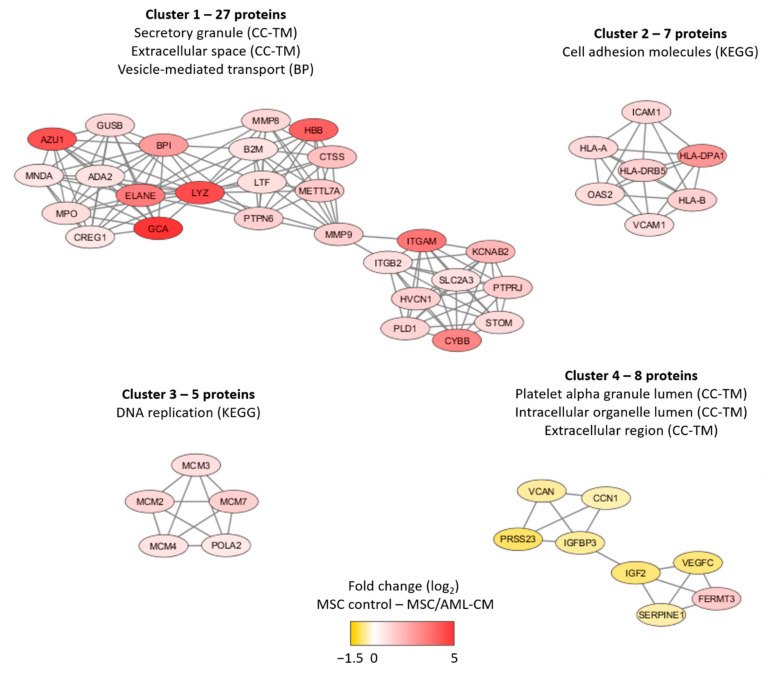
Subclusters of at least five densely connected proteins significantly altered after MSC exposure to AML conditioned medium. The subclusters were identified by the MCODE application in Cytoscape from a protein-protein interaction network analysis in String based on 201 MSC proteins that differed significantly when comparing MSCs exposed to AML conditioned media with MSCs derived from medium control cultures (see Appendix A for the overall network analysis results). The color coding of the protein nodes indicates the log_2_-transformed protein fold change, where yellow indicates decreased protein abundance and red indicates increased protein abundance in MSCs after exposure to AML conditioned medium relative to MSC control. The figure presents the most significant GO-terms/KEGG pathways for each network (see also Appendix A).

**Table 1 diseases-09-00074-t001:** Clinical and biological characteristics of the 41 AML patients. Unless otherwise stated the results are presented as the number of patients (for additional details see Appendix A).

Sex and Age (*n* = 41)		Karyotype/Cytogenetic Abnormalities	
Males/females	22/19	Normal	21
Age (years; median/range)	70/18–87	Favorable	4
		Intermediate	9
**Predisposition/previous disease**		Adverse	4
Previous chronic myeloid neoplasia	1	Not tested	3
Myelodysplastic syndrome	8		
Relapsed AML	3	**Flt3 status**	
Chemotherapy related	0	ITD	14
		Wild type	19
**Morphology/FAB classification**		Not tested	8
M0/M1	17		
M2	8	**NPM1 status**	
M4/M5	16	Insertion	14
M6/M7	0	Insertion + Flt3-ITD	9
		Wild type	20
**CD34 positive**	22	Not tested	7

Abbreviations: FAB, French-American-British; ITD, internal tandem duplication.

**Table 2 diseases-09-00074-t002:** Classification of the 217 MSC proteins that differed significantly when comparing the proteomic profiles of the two main patient clusters identified in the unsupervised hierarchical cluster analysis (Figure 2). The proteins were classified using a GO tool. All proteins in the statistical analysis were quantified for at least three patients in each of the two patient subsets, and 217 proteins showed a significant difference defined as *p* < 0.01 both for the *t*-test and for the Z-statistics of fold changes. The 10 terms with the lowest False Discovery Rate (FDR) are listed.

Term	Description	Category	Foreground Count ^1^	Background Count ^1^	s-Value ^1^	*p*-Value	FDR
**INCREASED IN THE UPPER YELLOW MAIN PATIENT CLUSTER (*n* = 10)**							
KW-0496	Mitochondrion	UniProt	57	833	1.53	1.57 × 10^−6^	0.00073
KW-0809	Transit peptide	UniProt	41	421	1.33	6.18 × 10^−7^	0.00073
KW-1274	Primary mitochondrial disease	UniProt	18	139	0.62	6.78 × 10^−7^	0.00073
KW-0276	Fatty acid metabolism	UniProt	11	98	0.29	1.32 × 10^−5^	0.0022
GO:0005739	Mitochondrion	GO CC	62	1065	1.60	9.00 × 10^−7^	0.0025
DOID:700	Mitochondrial metabolism disease	DO	18	138	0.62	6.24 × 10^−7^	0.0027
GOCC:0005739	Mitochondrion	CC-TM	54	954	1.30	1.73 × 10^−6^	0.0035
GOCC:0005759	Mitochondrial matrix	CC-TM	26	304	0.79	8.88 × 10^−7^	0.0035
DOID:3652	Leigh disease	DO	10	62	0.33	1.82 × 10^−6^	0.0039
KW-0732	Signal	UniProt	38	913	0.55	3.87 × 10^−5^	0.0045
**DECREASED IN THE UPPER YELLOW (i.e., INCREASED IN THE LOWER BROWN MAIN PATIENT CLUSTER)**							
KW-0176	Collagen	UniProt	7	31	0.74	4.39 × 10^−8^	5.16 × 10^−5^
KW-0732	Signal	UniProt	33	913	2.31	4.64 × 10^−7^	0.00027
KW-0325	Glycoprotein	UniProt	37	1180	2.29	1.22 × 10^−6^	0.00030
KW-1015	Disulfide bond	UniProt	29	880	1.87	8.93 × 10^−7^	0.00030
KW-0964	Secreted	UniProt	23	500	1.67	7.71 × 10^−7^	0.00030
KW-0272	Extracellular matrix	UniProt	13	103	1.07	1.04 × 10^−6^	0.00030
map04974	Protein digestion and absorption	KEGG	6	30	0.52	8.07 × 10^−7^	0.00035
GO:0030312	External encapsulating structure	CC	23	228	2.00	3.88 × 10^−7^	0.0011
GO:0031012	Extracellular matrix	CC	23	228	2.00	3.88 × 10^−7^	0.0011
GO:0005576	Extracellular region	CC	39	1884	1.92	9.28 × 10^−7^	0.0011

^1^ Foreground counts indicate the number of positive associations for each term (i.e., the number of proteins associated with the given term) and background counts indicate the number of positive associations in the dataset. S-value is a combination of (minus log) *p* value and effect size (i.e., positive associations in the foreground divided by all associations); a positive value indicates overrepresentation of a given term, and a negative value indicates underrepresentation of a given term. Abbreviations: BP, GO biological process; CC, GO Cellular component; CC-TM, GO Cellular component TextMining; DO, Disease Ontology; FDR, false discovery rate; KEGG, KEGG pathway; UniProt, Uniprot keyword.

**Table 3 diseases-09-00074-t003:** Classification of AML-induced MSC proteins (i.e., proteins quantified only for MSCs incubated with AML conditioned medium but not for control MSCs incubated in medium alone) using a GO tool. The three networks/subclusters presented in Figure 4 were analyzed separately. The most significant general GO terms and/or KEGG pathways are listed.

Term	Description	Category	Foreground Count ^1^	Background Count ^1^	s-Value ^1^	*p*-Value	FDR
**NETWORK 1**							
GO:0031410	Cytoplasmic vesicle	CC	20	1375	5.58	1.23 × 10^−7^	0.00034
GO:0005886	Plasma membrane	CC	20	1825	4.63	6.02 × 10^−7^	0.00083
GO:0016192	Vesicle-mediated transport	BP	20	1217	5.43	2.80 × 10^−7^	0.0028
GO:0002376	Immune system process	BP	20	1279	5.15	5.27 × 10^−7^	0.0028
GO:0006810	Transport	BP	20	2258	4.77	1.04 × 10^−7^	0.0021
GO:0007155	Cell adhesion	BP	9	372	2.59	3.17 × 10^−7^	0.0028
**NETWORK 2**							
GO:0005576	Extracellular region	CC	16	1873	3.85	2.27 × 10^−7^	0.00063
GO:0005615	Extracellular space	CC	15	1644	3.55	4.32 × 10^−7^	0.00063
GO:0031410	Cytoplasmic vesicle	CC	14	1375	3.45	4.54 × 10^−7^	0.00063
GO:0016192	Vesicle-mediated transport	BP	17	1217	5.08	9.71 × 10^−8^	0.0019
GO:0006950	Response to stress	BP	15	1705	3.38	7.15 × 10^−7^	0.0027
GO:0040011	Locomotion	BP	10	515	3.03	2.14 × 10^−7^	0.0019
GO:0048870	Cell motility	BP	9	435	2.55	6.77 × 10^−7^	0.0027
map04060	Cytokine-cytokine receptor interaction	KEGG	7	47	2.41	2.20 × 10^−7^	9.60 × 10^−5^
map04062	Chemokine signaling pathway	KEGG	7	102	2.34	2.54 × 10^−7^	9.60 × 10^−5^
**NETWORK 3**							
GO:0005886	Plasma membrane	CC	13	1825	4.47	2.27 × 10^−7^	0.00063
GO:0007165	Signal transduction	BP	12	1753	3.40	2.77 × 10^−6^	0.028
map04666	Fc gamma R-mediated phagocytosis	KEGG	5	72	2.30	2.38 × 10^−7^	0.0001
map04062	Chemokine signaling pathway	KEGG	5	102	2.04	1.24 × 10^−6^	0.00014

^1^ Foreground counts indicate the number of positive associations for each term (i.e., the number of proteins associated with the given term) and background counts indicate the number of positive associations in the dataset. S-value is a combination of (minus log) *p* value and effect size (i.e., positive associations in the foreground divided by all associations); a positive value indicates overrepresentation of a given term, and a negative value indicates underrepresentation of a given term. Abbreviations: BP, GO biological process; CC, FDR, false discovery rate; GO cellular component; KEGG, KEGG pathway.

## Data Availability

The data presented in this study are available on request from the corresponding author.

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
