# Peer review of "Patient Heterogeneity in Acute Myeloid Leukemia: Leukemic Cell Communication by Release of Soluble Mediators and Its Effects on Mesenchymal Stem Cells"

_diseases, 2021, doi:10.3390/diseases9040074_

Round 1
Reviewer 1 Report
The manuscript is overall well performed and statistically sound. The “Introduction” and “Materials and Methods” sections are short and appropriately referenced. The “Results” section describes the data presented in the tables and figures. The results are quite convincing. The “Discussion” paragraph clarifies the findings. This key section is carefully organized and enhances the readability of the manuscript. Overall the manuscript is well performed and statistically sound.
Minor concerns:
Table legends are quite long and should be shortened. “Table titles” should be placed above the table. Any detailed information can be given below the table.
Author Response
The manuscript is overall well performed and statistically sound. The “Introduction” and “Materials and Methods” sections are short and appropriately referenced. The “Results” section describes the data presented in the tables and figures. The results are quite convincing. The “Discussion” paragraph clarifies the findings. This key section is carefully organized and enhances the readability of the manuscript. Overall the manuscript is well performed and statistically sound.
Minor concerns:
Table legends are quite long and should be shortened. “Table titles” should be placed above the table. Any detailed information can be given below the table.
Response: As suggested by the reviewer we have shortened the text in the headings of Tables 2, 3 and S9 (previously Table 4). The explanation of the various headings/columns is now presented below the table as suggested by the reviewer together with the list of abbreviations. The same has been done for Tables S8 and S9 in the Supplementary information.
Reviewer 2 Report
The authors described the results of experiments on leukemic cell communication by release of soluble mediators and its effects on mesenchymal stem cells (MSC). They concluded that the AML-modulated MSC proteins formed several interacting networks mainly reflecting intracellular organellar structure/trafficking but also extracellular matrix/cytokine signaling, and a single small network reflecting altered DNA replication. The results that extracellular mediator release by primary human AML cells alters proteomic profiles of normal bone marrow MSCs and the authors’ interpretations are intriguing. However, there are several issues to be corrected or clarified.
Majors)
1) The discrepancy as described in below should be corrected.
In Table 1, “Chemotherapy related” is described as “0”. In the text (Page 7, Line245~ and 255~), however, “All these three patients were elderly males with secondary AML, 245 but they differed with regard to differentiation (i.e. FAB classification, CD34 expression) 246 and genetic abnormalities (Table S1).” and “This upper/yellow main patient cluster had a significantly higher frequency of patients with secondary AML (Fisher’s exact test, p=0.0165),”
2) Page 10, Line 255~) “3.5. A subset of MSC proteins can only be quantified after exposure to the AML secretome but even these proteins also contribute to AML heterogeneity”
In contrast, how about MSC proteins which could be quantified only before exposure to the AML conditioned medium?
3) Page 17, Line 556~) “Finally, due to this consecutive selection a large subset of our patients was elderly or unfit, and only a minority of our patients completed potentially curative intensive treatment [1]. For these reasons, in our opinion, the present study included too few patients for survival analyses.”
Results of an exploratory analysis with and without relapsed patients on survival should be added at least in Supplements.
Minors)
1) This manuscript is relatively long with many repetitions.
For example, “S-value is a combination of (minus log) p value and effect size (i.e. positive associations in the foreground divided by all associations); a positive value indicates overrepresentation of a given term, and a negative value indicates underrepresentation of a given term. Foreground counts indicate the number of positive associations in a given cluster for a given term (i.e. the number of proteins associated with the given term) and background counts indicate the number of positive association in the dataset.” is described in both Table 3 and Table 4. Text should be condensed.
As shown above are also applicable for Supplements.
Also, about 20% of the total amount of the Text, Tables and Figures should be transferred to the Supplements.
2) Limitations of this manuscript should be summarized in brief in Discussion.
Author Response
The authors described the results of experiments on leukemic cell communication by release of soluble mediators and its effects on mesenchymal stem cells (MSC). They concluded that the AML-modulated MSC proteins formed several interacting networks mainly reflecting intracellular organellar structure/trafficking but also extracellular matrix/cytokine signaling, and a single small network reflecting altered DNA replication. The results that extracellular mediator release by primary human AML cells alters proteomic profiles of normal bone marrow MSCs and the authors’ interpretations are intriguing. However, there are several issues to be corrected or clarified.
Majors)
1) The discrepancy as described in below should be corrected.
In Table 1, “Chemotherapy related” is described as “0”. In the text (Page 7, Line245~ and 255~), however, “All these three patients were elderly males with secondary AML, 245 but they differed with regard to differentiation (i.e. FAB classification, CD34 expression) 246 and genetic abnormalities (Table S1).” and “This upper/yellow main patient cluster had a significantly higher frequency of patients with secondary AML (Fisher’s exact test, p=0.0165),”
Response: We apologize for this. This part has now been rewritten; it is clearly stated that this patient subset showed an increased frequency of patients with “AML following previous myelodysplastic syndrome or chronic myeloproliferative disease” (page 7).
2) Page 10, Line 255~) “3.5. A subset of MSC proteins can only be quantified after exposure to the AML secretome but even these proteins also contribute to AML heterogeneity”
In contrast, how about MSC proteins which could be quantified only before exposure to the AML conditioned medium?
Response: We have now done these additional analyses. The results are presented in a new Section 3.7, a detailed description of the identified proteins is given in the new Table S10 and the observations are briefly commented in the Discussion section (page 17).
3) Page 17, Line 556~) “Finally, due to this consecutive selection a large subset of our patients was elderly or unfit, and only a minority of our patients completed potentially curative intensive treatment [1]. For these reasons, in our opinion, the present study included too few patients for survival analyses.”
Results of an exploratory analysis with and without relapsed patients on survival should be added at least in Supplements.
Response: We apologize that we did not explain the basis for our statement that survival analyses could not be performed. We have now included more detailed information about the treatment and the survival of each individual patient in Table S1. It can be seen that our patients could be classified into three main subsets based on the treatment:
- Twenty patients started intensive antileukemic treatment, but three patients had toxic death and one patient denied to complete the antileukemic treatment. Thus, only 16 patients completed the planned treatment, i.e. the efficiency of a relevant/completed antileukemic treatment can only be evaluated for 16 patients, and as can be seen from the table we had only four long-term AML-free survivors among these 16 patients. The survival is presented as survival until nonrelapse/relapse death, or as the time of follow-up for the AML-free survivors.
- Five patients received only AML-stabilizing treatment; all these patients died from relapse after relatively short duration of treatment.
- Sixteen patients received only supportive therapy (including three patients with relapse), and all these patients showed an expected short survival, i.e. less than three months.
In the present context the intention of a survival analysis should be to investigate whether identified patient subsets differ in their clinical chemosensitivity, i.e. their survival. Furthermore, in our opinion an overall survival analysis should then be based on patients receiving a comparable treatment, i.e. patients completing the planned intensive treatment possibly including stem cell transplantation. Patients with early toxic death should then be ledt out together with patients not receiving or completing the planned overall treatment, because in our present context inclusion of such patients in the survival analysis would disturb the intention of the survival analysis, i.e. to investigate whether patients subsets identified by analysis of the MSC proteomic profile differ in chemosensitivity/survival. Thus, for patients receiving intensive treatment one should only include the 16 patients that completed the treatment, because the intention would then be to compare the antileukemic efficiency for two different patient subset (i.e. the two main subsets). It is not possible to know the efficiency of a relevant treatment for patients with early toxic death or patients refusing to complete the treatment. A comparison of survival after intensive treatment for the smallest main patient subset (Figure 2, Table S1 patients 1-10) versus the larger main cluster (patients 11-41) then has to be a comparison of survival for four patients versus 12 patients. In our opinion these groups are too small for a reliable comparison. All the patients in the two other treatment groups showed very short survival.
We hope this can be accepted. Our priority was to investigate a representative group of consecutive/unselected patients, thus not only including younger fit patients but patients at all ages. Our study should therefore in our opinion be regarded as representative, but by using this strategy approximately half of the patients will (as expected) be elderly or unfit patients that cannot receive potentially curative treatment.
We hope our explanation can be accepted. We have now included detailed information about treatment and survival in Table S1, and a brief explanation is added in the Discussion section (page 16).
Minors)
1) This manuscript is relatively long with many repetitions.
For example, “S-value is a combination of (minus log) p value and effect size (i.e. positive associations in the foreground divided by all associations); a positive value indicates overrepresentation of a given term, and a negative value indicates underrepresentation of a given term. Foreground counts indicate the number of positive associations in a given cluster for a given term (i.e. the number of proteins associated with the given term) and background counts indicate the number of positive association in the dataset.” is described in both Table 3 and Table 4. Text should be condensed.
As shown above are also applicable for Supplements.
Response: The table headings have been simplified as suggested also by the other reviewer. We have also shortened the text as described below.
We agree that our tables in the Supplementary information are extensive. However, we hope that our article will be of interest for clinicians, and especially clinicians investigating targeted or personalized AML therapy. Several new approaches now become available, and these new therapeutic strategies are often targeting defined molecules, molecular complexes of signaling pathways. Our intention has been to make molecular details easily available for the reader through the protein lists presented in the Supplementary information. This information will then be available without any disturbance of the descriptions/presentations of our main observations in the Results section of our article, but it will be possible to get more detailed information without going through or analyzing large files with original data. However, we have revised the table legend/headings for Tables S8 and S9 (see comment from reviewer 1).
We agree that the legends in the Supplementary information are relatively extensive. We have done it in this way so that the reader will have all necessary information available in the supplementary information to understand the table; it should not be necessary to go back to find details in the article to understand the tables in the Supplementary section. We hope this can be accepted.
We would be grateful if we could keep the information in the Supplementary information.
Also, about 20% of the total amount of the Text, Tables and Figures should be transferred to the Supplements.
Response: As suggested by the reviewer we have shortened our manuscript, the alterations are described more in detail below.
We have shortened the Introduction from 486 to 364 words.
The Material and methods section has been shortened from 1275 to 1135 words.
The text of the Results section has not been shortened, we hope this can be accepted. However, we have removed one figure and one table from the article, and they are now included in the Supplementary Information. As suggested by the reviewer the original Figure 3 is now transferred to the Supplementary information (new Figure S3). The original Table 4 has also been transferred to the Supplementary section (Table S9).
We have shortened the Discussion/Conclusion sections; the original parts of these two sections included 1624 words whereas the two present versions include 1418 words corresponding to a shortening of 206 words. However, due to the reviewer’s comment on survival analyses (see above), additional proteomic analyses and comment on study limitations we had to include three new statement corresponding to 27 (page 16), 34 words (page 17) and 47 words (pages 15-16), respectively; the original parts of these two sections have therefore been shortened by 314 words (206+27+34+47 words).
We hope our solutions can be accepted.
2) Limitations of this manuscript should be summarized in brief in Discussion.
Response: As suggested by the reviewer we briefly comment the major limitations of our study in the first chapter of the Discussion section (pages 15-16).
Round 2
Reviewer 2 Report
I think the response to the reviewers' comments and revision of the manuscripts are plausible.